# Sequential Experimental Design for Transductive Linear Bandits

**Tanner Fiez**
Electrical & Computer Engineering
University of Washington
fiezt@uw.edu

**Lalit Jain**[*]
Allen School of Computer Science & Engineering
University of Washington
lalitj@cs.washington.edu

**Kevin Jamieson**
Allen School of Computer Science & Engineering
University of Washington
jamieson@cs.washington.edu

**Lillian Ratliff**
Electrical & Computer Engineering
University of Washington
ratliffl@uw.edu

## Abstract

In this paper we introduce the pure exploration *transductive linear bandit problem*: given a set of measurement vectors $\mathcal{X} \subset \mathbb{R}^d$, a set of items $\mathcal{Z} \subset \mathbb{R}^d$, a fixed confidence $\delta$, and an unknown vector $\theta^* \in \mathbb{R}^d$, the goal is to infer $\operatorname{argmax}_{z \in \mathcal{Z}} z^\top \theta^*$ with probability $1 - \delta$ by making as few sequentially chosen noisy measurements of the form $x^\top \theta^*$ as possible. When $\mathcal{X} = \mathcal{Z}$, this setting generalizes *linear bandits*, and when $\mathcal{X}$ is the standard basis vectors and $\mathcal{Z} \subset \{0, 1\}^d$, *combinatorial bandits*. The transductive setting naturally arises when the set of measurement vectors is limited due to factors such as availability or cost. As an example, in drug discovery the compounds and dosages $\mathcal{X}$ a practitioner may be willing to evaluate in the lab in vitro due to cost or safety reasons may differ vastly from those compounds and dosages $\mathcal{Z}$ that can be safely administered to patients in vivo. Alternatively, in recommender systems for books, the set of books $\mathcal{X}$ a user is queried about may be restricted to known best-sellers even though the goal might be to recommend more esoteric titles $\mathcal{Z}$. In this paper, we provide instance-dependent lower bounds for the transductive setting, an algorithm that matches these up to logarithmic factors, and an evaluation. In particular, we present the first non-asymptotic algorithm for linear bandits that nearly achieves the information-theoretic lower bound.

## 1 Introduction

In content recommendation or property optimization in the physical sciences, often there is a set of items (e.g., products to purchase, drugs) described by a set of feature vectors $\mathcal{Z} \subset \mathbb{R}^d$, and the goal is to find the $z \in \mathcal{Z}$ that maximizes some response or property (e.g., affinity of user to the product, drug combating disease). A natural model for these settings is to assume that there is an unknown vector $\theta^* \in \mathbb{R}^d$ and the expected response to any item $z \in \mathcal{Z}$, if evaluated, is equal to $z^\top \theta^*$. However, we often cannot measure $z^\top \theta^*$ directly, but we may infer it transductively through some potentially noisy probes. That is, given a finite set of probes $\mathcal{X} \subset \mathbb{R}^d$ we observe $x^\top \theta^* + \eta$ for any $x \in \mathcal{X}$ where $\eta$ is independent mean-zero, sub-Gaussian noise. Given a set of measurements $\{(x_i, r_i)\}_{i=1}^N$ one can construct the least squares estimator $\widehat{\theta} = \arg\min_\theta \sum_{i=1}^N (r_i - x_i^\top \theta)^2$ and then use $\widehat{\theta}$ as a plug-in estimate for $\theta^*$ to estimate the optimal $z_* := \operatorname{argmax}_{z \in \mathcal{Z}} z^\top \theta^*$. However, the accuracy of such a plug-in estimator depends critically on the number and choice of probes used to construct

---

[*]Contribution shared equally among T. Fiez and L. Jain.

$\widehat{\theta}$. Unfortunately, the optimal allocation of probes cannot be decided a priori: it must be chosen sequentially and adapt to the observations in real-time to optimize the accuracy of the prediction.

If the probing vectors (arms) $\mathcal{X}$ are *equal* to the item vectors $\mathcal{Z}$, this problem is known as *pure exploration for linear bandits* which is considered in [21, 30, 31, 33]. This naturally arises in content recommendation, for example, if $\mathcal{X} = \mathcal{Z}$ is a feature representation of songs, and $\theta^*$ represents a user's music preferences, a music recommendation system can elicit the preference for a particular song $z \in \mathcal{Z}$ directly by enqueuing it in the user's playlist. However, often times there are constraints on which items in $\mathcal{Z}$ can be shown to the user.

1. $\mathcal{X} \subset \mathcal{Z}$. Consider a whiskey bar with hundreds of whiskies ranging in price from dollars a shot to hundreds of dollars. The bar tender may have an implicit feature representation of each whiskey, the patron has an implicit preference vector $\theta^*$, and the bar tender wants to select the affordable whiskeys $\mathcal{X} \subset \mathcal{Z}$ in a taste test to get an idea of the patron's preferences before recommending the expensive whiskies that optimize the patron's preferences in $\mathcal{Z}$.
2. $\mathcal{Z} \subset \mathcal{X}$. In drug discovery, thousands of compounds are evaluated in order to determine which ones are effective at combating a disease. However, it may be that while $\mathcal{Z}$ is the set of compounds and doses that are approved for medical use (e.g., safe), it may be advantageous to test even unsafe compounds or dosages $\mathcal{X}$ such that $\mathcal{X} \supset \mathcal{Z}$. Such unsafe $\mathcal{X}$ may aid in predicting the optimal $z_* \in \mathcal{Z}$ because they provide more information about $\theta^*$.
3. $\mathcal{Z} \cap \mathcal{X} = \emptyset$. Consider a user shopping for a home among a set $\mathcal{Z}$ where each is parameterized by a number of factors like distance to work, school quality, crime rate, etc. so that each $z \in \mathcal{Z}$ can be described as a linear combination of the relevant factors described by $\mathcal{X}$: $z = \sum_{x \in \mathcal{X}} \alpha_{z,x} x$, where we may take each $x \in \mathcal{X}$ to simply be one-hot-encoded. The response $x^\top \theta^* + \eta$ reflects the user's preferences for the query $x$, a specific attribute of the house. Indeed, if all $\alpha_{z,x} \in \{0, 1\}$ this is known as *pure exploration for combinatorial bandits* [10, 8]. That is, a house either has the attribute, or not.

Given items $\mathcal{Z}$, measurement probes $\mathcal{X}$, a confidence $\delta$, and an unknown $\theta^*$, this paper develops algorithms to sequentially decide which measurements in $\mathcal{X}$ to take in order to minimize the number of measurements necessary in order to determine $z_*$ with high probability.

## 1.1 Contributions

Our goals are broadly to first define the transductive bandit problem and then characterize the instance-optimal sample complexity for this problem. Our contributions include the following.

1. In Section 2 we provide instance dependent lower bounds for the transductive bandit problem that simultaneously generalize previous known lower bounds for linear bandits and combinatorial bandits using standard arguments.
2. In Section 3 we give an algorithm (Algorithm 1) for transductive linear bandits and prove an associated sample complexity result (Theorem 2). We show that the sample complexity we obtain matches the lower bound up to logarithmic factors. This is the primary contribution of the paper. Along the way, we discuss how rounding procedures can be used to improve upon the computational complexity of this algorithm.
3. In Sections 4 and 5 we contrast our algorithm with previous work from a theoretical and empirical perspective, respectively. Our experiments show that our theoretically superior algorithm is empirically competitive with previous algorithms on a range of problem scenarios.

## 1.2 Notation

For each $z \in \mathcal{Z}$ define the *gap* of $z$, $\Delta(z) = (z_* - z)^\top \theta^*$ and furthermore, $\Delta_{\min} = \min_{z \neq z_*} \Delta(z)$. If $A \in \mathbb{R}^{d \times d}_{\geq 0}$ is a positive semidefinite matrix, and $y \in \mathbb{R}^d$ is a vector, let $\|y\|_A^2 := y^\top A y$ denote the induced semi-norm. Let $\triangle_{\mathcal{X}} := \{\lambda \in \mathbb{R}^{|\mathcal{X}|} : \lambda \geq 0, \sum_{x \in \mathcal{X}} \lambda_x = 1\}$ denote the set of probability distributions on $\mathcal{X}$. Taking $\mathcal{S} \subset \mathcal{Z}$ to a subset of the arm set, we define two operators we define $\mathcal{Y}(\mathcal{S}) = \{z - z' : \forall z, z' \in \mathcal{S}, z \neq z'\}$ as the directions obtained from the differences between each pair of arms and $\mathcal{Y}^*(\mathcal{S}) = \{z_* - z : \forall z \in \mathcal{S} \setminus z_*\}$ as the directions obtained from the differences between the optimal arm and each suboptimal arm. Finally, for an arbitrary set of vectors $\mathcal{V} \subset \mathbb{R}^d$, define $\rho(\mathcal{V}) = \min_{\lambda \in \triangle_{\mathcal{X}}} \max_{v \in \mathcal{V}} \|v\|^2_{(\sum_{x \in \mathcal{X}} \lambda_x x x^\top)^{-1}}$. This quantity will be crucial in the discussion of our sample complexity and it is motivated in Section 2.2

## 2 Transductive Linear Bandits Problem

Consider known finite collections of $d$-dimensional vectors $\mathcal{X} \subset \mathbb{R}^d$ and $\mathcal{Z} \subset \mathbb{R}^d$, known confidence $\delta \in (0,1)$, and unknown $\theta^* \in \mathbb{R}^d$. The objective is to identify $z_* = \mathrm{argmax}_{z \in \mathcal{Z}} z^\top \theta^*$ with probability at least $1 - \delta$ while taking as few measurements in $\mathcal{X}$ as possible. Formally, a transductive linear bandits algorithm is described by a **selection rule** $X_t \in \mathcal{X}$ at each time $t$ given the history $(X_s, R_s)_{s<t}$, **stopping time** $\tau$ with respect to the filtration $\mathcal{F}_t = (X_s, R_s)_{s \leq t}$, and **recommendation rule** $\widehat{z} \in \mathcal{Z}$ invoked at time $\tau$ which is $\mathcal{F}_\tau$-measurable. We assume that $X_t$ is $\mathcal{F}_{t-1}$-measurable and may use additional sources of randomness; in addition at each time $t$ that $R_t = X_t^\top \theta^* + \eta_t$ where $\eta_t$ is independent, zero-mean, and 1-sub-Gaussian. Let $\mathbb{P}_{\theta^*}, \mathbb{E}_{\theta^*}$ denote the probability law of $R_t | \mathcal{F}_{t-1}$ for all $t$.

**Definition 1.** *We say that an algorithm for a transductive bandit problem is $\delta$-PAC for $\mathcal{X}, \mathcal{Z} \subset \mathbb{R}^d$ if for all $\theta^* \in \mathbb{R}^d$ we have $\mathbb{P}_{\theta^*}(\widehat{z} = z_*) \geq 1 - \delta$.*

### 2.1 Optimal allocations

In this section we discuss a number of ways we can allocate a measurement budget to the different arms. The following establishes a lower bound on the expected number of samples any $\delta$-PAC algorithm must take.

**Theorem 1.** *Assume $\eta_t \overset{iid}{\sim} \mathcal{N}(0,1)$ for all $t$. Then for any $\delta \in (0,1)$, any $\delta$-PAC algorithm must satisfy*

$$\mathbb{E}_{\theta^*}[\tau] \geq \log(1/2.4\delta) \min_{\lambda \in \triangle_{\mathcal{X}}} \max_{z \in \mathcal{Z} \setminus \{z_*\}} \frac{\|z_* - z\|^2_{(\sum_{x \in \mathcal{X}} \lambda_x x x^\top)^{-1}}}{((z_* - z)^\top \theta^*)^2}.$$

This lower bound is proved in Appendix C using standard techniques and employs the transportation inequality of [22]. It generalizes a previous lower bound in the setting of linear bandits [29] and lower bounds in the combinatorial bandit literature [10].

**Optimal static allocation.** To demonstrate that this lower bound is tight, define

$$\lambda^* := \underset{\lambda \in \triangle_{\mathcal{X}}}{\mathrm{argmin}} \max_{z \in \mathcal{Z} \setminus \{z_*\}} \frac{\|z_* - z\|^2_{(\sum_{x \in \mathcal{X}} \lambda_x x x^\top)^{-1}}}{((z_* - z)^\top \theta^*)^2} \text{ and } \psi^* = \max_{\mathcal{Z} \setminus \{z_*\}} \frac{\|z_* - z\|^2_{(\sum_{x \in \mathcal{X}} \lambda_x^* x x^\top)^{-1}}}{((z_* - z)^\top \theta^*)^2},$$
(1)

where $\psi^*$ is the value of the lower bound and $\lambda^*$ is the allocation that achieves it. Suppose we sample arm $x \in \mathcal{X}$ exactly $2\lfloor \lambda_x^* N \rfloor$ times where we assume[2] $N \in \mathbb{N}$ is sufficiently large so that $\min_{x:\lambda_x>0} \lfloor \lambda_x N \rfloor > 0$. If $N = \lceil 2\psi^* \log(|\mathcal{Z}|/\delta) \rceil$ then as we will show shortly (Section 2.2), the least squares estimator $\widehat{\theta}$ satisfies $(z_* - z)^\top \widehat{\theta} > 0$ for all $z \in \mathcal{Z} \setminus z_*$ with probability at least $1 - \delta$. Thus, with probability at least $1 - \delta$, $z_*$ is equal to $\widehat{z} = \arg\max_{z \in \mathcal{Z}} z^\top \widehat{\theta}$ and the total number of samples is bounded by $2N$ which is within $4 \log(|\mathcal{Z}|)$ of the lower bound. Unfortunately, of course, the allocation $\lambda^*$ relies on knowledge of $\theta^*$ (which determines $z_*$) which is unknown a priori, and thus this is not a realizable strategy.

**Other static allocations.** Short of $\lambda^*$ it is natural to consider allocations that arise from optimal linear experimental design [27]. For the special case of $\mathcal{X} = \mathcal{Z}$ it has been argued ad nauseam that a $G$-optimal design, $\mathrm{argmin}_{\lambda \in \triangle_{\mathcal{X}}} \max_{x \in \mathcal{X}, x \neq x_*} \|x\|^2_{(\sum_{x \in \mathcal{X}} \lambda_x x x^\top)^{-1}}$, is woefully loose since it does not utilize the differences $x - x'$, $x, x' \in \mathcal{X}$ [25, 30, 33]. Also for the $\mathcal{X} = \mathcal{Z}$ case, [34, 30] have proposed the static $\mathcal{X}\mathcal{Y}$-allocation given as $\mathrm{argmin}_{\lambda \in \triangle_{\mathcal{X}}} \max_{x,x' \in \mathcal{X}} \|x - x'\|^2_{(\sum_{x \in \mathcal{X}} \lambda_x x x^\top)^{-1}}$. In [30] it is shown that no more than $O(\frac{d}{\Delta^2_{\min}} \log(|\mathcal{X}| \log(1/\Delta_{\min})/\delta))$ samples from each of these allocations suffice to identify the best arm. While the above discussion demonstrates that for every $\theta^*$ there exists an optimal static allocation (that explicitly uses $\theta^*$) nearly achieving the lower bound, any static allocation with no prior knowledge of $\theta^*$ can require a factor of $d$ more samples than necessary.

**Proposition 1.** *Let $c, c'$ be universal constants. For any $\gamma > 0$, $d$ even, there exists sets $\mathcal{X} = \mathcal{Z} \subset \mathbb{R}^d$ and a set $\Theta \subset \mathbb{R}^d$, such that $\inf_{\mathcal{A}} \max_{\theta \in \Theta} \mathbb{E}_\theta[\tau] \geq \frac{cd \log(1/\delta)}{\gamma}$ where $\mathcal{A}$ is the set of all algorithms that are $\delta$-PAC for $\mathcal{X}, \mathcal{Z}$ and take a static allocation of samples. On the other hand $\psi^*/c' \leq d + \frac{1}{\gamma}$ for every choice of $\theta^* \in \Theta$.*

This proposition indicates that it is necessary to devise an adaptive algorithm to obtain a instance-optimal sample complexity. The proof of this proposition can be found in Appendix D.

**Adaptive allocations.** As suggested by the problem definition, our strategy is to adapt the allocation over time, informed by the observations up to the current time. Specifically, our algorithm will proceed in rounds where at round $t$, we perform an $\mathcal{X}\mathcal{Y}$-allocation that is sufficient to remove all arms $z \in \mathcal{Z}$ that have gaps of at least $2^{-(t-1)}$. We show that the total number of measurements accumulates to $\psi^* \log(|\mathcal{Z}|^2/\delta)$ times some additional logarithmic factors, nearly achieving the optimal allocation as well as the lower bound. In Section 4, we review related procedures for the specific case of $\mathcal{X} = \mathcal{Z}$.

## 2.2 Review of Least Squares

Given a fixed design $\mathbf{x}_T = (x_t)_{t=1}^T$ with each $x_t \in \mathcal{X}$ and associated rewards $(r_t)_{t=1}^T$, a natural approach is to construct the ordinary-least squares (OLS) estimate $\widehat{\theta} = (\sum_{t=1}^T x_t x_t^\top)^{-1}(\sum_{t=1}^T r_t x_t)$. One can show $\widehat{\theta}$ is unbiased with covariance $\preceq (\sum_{t=1}^T x_t x_t^\top)^{-1}$. Moreover, for any $y \in \mathbb{R}^d$, we have[3]

$$\mathbb{P}\left(y^\top(\theta^* - \widehat{\theta}) \geq \sqrt{\|y\|^2_{(\sum_{t=1}^T x_t x_t^\top)^{-1}} 2\log(1/\delta)}\right) \leq \delta. \tag{2}$$

In particular, if we want this to hold for all $y \in \mathcal{Y}^*(\mathcal{Z})$, we need to union bound over $\mathcal{Z}$ replacing $\delta$ with $\delta/|\mathcal{Z}|$. Let us now use this to analyze the procedure discussed above (in the discussion on the optimal static allocation after Theorem 1) that gives an allocation matching the lower bound. With the choice of $N = \lceil 2\psi^* \log(|\mathcal{Z}|/\delta)\rceil$ and the allocation $2\lfloor \lambda_x^* N\rfloor$ for each $x \in \mathcal{X}$, we have for each $z \in \mathcal{Z} \setminus z_*$ that with probability at least $1 - \delta$,

$$(z_* - z)^\top \widehat{\theta} \geq (z_* - z)^\top \theta^* - \sqrt{\|z_* - z\|^2_{(\sum_x 2\lfloor N\lambda_x^* \rfloor xx^T)^{-1}} 2\log(|\mathcal{Z}|/\delta)} \geq 0$$

since for each $y = z_* - z \in \mathcal{Y}^*(\mathcal{Z})$ we have

$$y^\top \left(\sum_{x \in \mathcal{X}} 2\lfloor N\lambda_x^*\rfloor xx^\top\right)^{-1} y \leq y^\top \left(\sum_{x \in \mathcal{X}} \lambda_x^* xx^\top\right)^{-1} y/N \leq ((z_* - z)^\top \theta^*)^2/(2\log(|\mathcal{Z}|/\delta)), \tag{3}$$

where the last inequality plugs in the value of $N$ and the definition of $\psi^*$. The fact that at most one $z' \in \mathcal{Z}$ can satisfy $(z' - z)^\top \widehat{\theta} > 0$ for all $z \neq z' \in \mathcal{Z}$, and that $z' = z_*$ does, certifies that $\widehat{z} = \arg\max_{z \in \mathcal{Z}} z^\top \widehat{\theta}$ is indeed the best arm with probability at least $1 - \delta$. Note that equation (3) provides the motivation for how the form of $\psi^*$ is obtained. Rearranging, it is equivalent to,

$$N \geq 2\log(|\mathcal{Z}|/\delta) \max_{\mathcal{Z}\setminus\{z_*\}} \frac{\|z_* - z\|^2_{(\sum_{x \in \mathcal{X}} \lambda_x^* xx^\top)^{-1}}}{((z_* - z)^\top \theta^*)^2} \text{ for all } z \in \mathcal{Z} \setminus \{z_*\}$$

Thinking of the right hand side of the inequality as a function of $\lambda$, $\lambda^*$ is precisely chosen to minimize this quantity and hence the sample complexity.

## 2.3 Rounding Procedures

We briefly digress to address a technical issue. Given an allocation $\lambda$ and an arbitrary subset of vectors $\mathcal{Y}$, in general, drawing $N$ samples $\mathbf{x}_N := \{x_1, \ldots, x_N\}$ at random from $\mathcal{X}$ according to the distribution $\lambda_x$ may result in a design where $\max_{y \in \mathcal{Y}} \|y\|^2_{(\sum_{t=1}^N x_t x_t^\top)^{-1}}$ (which appears in the width of the confidence interval (2)) differs significantly from $\max_{y \in \mathcal{Y}} \|y\|^2_{(\sum_{x \in \mathcal{X}} \lambda_x xx^\top)^{-1}}/N$. Naive strategies for choosing $\mathbf{x}_N$ will fail. We can not simply use an allocation of $N\lambda_x$ samples for any specific $x$ since this may not be an integer. Furthermore, greedily rounding $N\lambda_x$ to an allocation $\lfloor N\lambda_x \rfloor$ or $\lceil N\lambda_x \rceil$ may result in fewer than necessary, or far more than $N$ total samples if the support of $\lambda$ is large. However, given $\epsilon > 0$, there are *efficient rounding procedures* that produce $(1 + \epsilon)$ approximations as long as $N$ is greater than some minimum number of samples $r(\epsilon)$. In short, given $\lambda$ and a choice of $N$ they return an allocation $\mathbf{x}_N$ satisfying $\max_{y \in \mathcal{Y}} \|y\|^2_{(\sum_{i=1}^N x_i x_i^\top)^{-1}} \leq (1 + \epsilon)\max_{y \in \mathcal{Y}} \|y\|^2_{(\sum_{x \in \mathcal{X}} \lambda_x xx^\top)^{-1}}/N$. Such a procedure with $r(\epsilon) \leq O(d/\epsilon^2)$ is described in

Section B in the supplementary. In our experiments we use a rounding procedure from [27] that is easier to implement with $r(\epsilon) = 2\|\lambda\|_0/\epsilon \le (d(d+1)+2)/\epsilon$. In general, $\epsilon$ should be thought of as a constant. The number of samples $N$ we need to take in our algorithm will be significantly larger than $r(\epsilon)$, so the impact of the rounding procedure is minimal. We provide details on this rounding procedure in Section B of the supplementary (also see [30, Appendix C]).

## 3   Sequential Experimental Design for Transductive Linear Bandits

Our algorithm for the pure exploration transductive bandit is presented in Algorithm 1. The algorithm proceeds in rounds, keeping track of the active arms $\widehat{\mathcal{Z}}_t \subseteq \mathcal{Z}$ in each round $t$. At the start of round $t$, the algorithm samples in such a way to remove all arms with gaps greater than $2^{-(t-1)}$. Thus denoting $\mathcal{S}_t := \{z \in \mathcal{Z} : \Delta(z) \le 4 \cdot 2^{-t}\}$, in round $t$ we expect $\widehat{\mathcal{Z}}_t \subset \mathcal{S}_t$.

As described above, if we knew $\theta^*$, we would sample according to the optimal allocation $\arg\min_{\lambda \in \triangle_{\mathcal{X}}} \max_{z \in \widehat{\mathcal{Z}}_t} \|z_* - z\|^2_{(\sum_{x \in \mathcal{X}} \lambda_x x x^\top)^{-1}} / ((z_* - z)^\top \theta^*)^2$. However, if at the start of the round we only have an upper bound on the gaps $\Delta(z) \le 4 \cdot 2^{-t}$ and do not know $z^*$, we can use the triangle inequality to obtain $4 \max_{z \in \widehat{\mathcal{Z}}_t} \|z_* - z\|^2_{(\sum_{x \in \mathcal{X}} \lambda_x x x^\top)^{-1}} \ge \max_{y \in \mathcal{Y}(\widehat{\mathcal{Z}}_t)} \|y\|^2_{(\sum_{x \in \mathcal{X}} \lambda_x x x^T)^{-1}}$ and lower-bound the objective by $(2^{t-3})^2 \min_{\lambda \in \triangle_{\mathcal{X}}} \max_{y \in \mathcal{Y}(\widehat{\mathcal{Z}}_t)} \|y\|^2_{(\sum_{x \in \mathcal{X}} \lambda_x x x^T)^{-1}}$.[4] This motivates our choice of $\lambda_t$ and $\rho(\mathcal{Y}(\widehat{\mathcal{Z}}_t))$. Thus by the same logic used in Section 2.2, $N_t = \lceil 2(2^t)^2(1+\epsilon)\rho(\mathcal{Y}(\widehat{\mathcal{Z}}_t)) \log(|\mathcal{Z}|^2/\delta_t) \rceil$ samples should suffice to guarantee that we can construct a confidence interval on each $(z - z')^\top \theta^*$ for $(z - z') \in \mathcal{Y}(\widehat{\mathcal{Z}}_t)$ of size at most $2^{-t}$ (with the $|\mathcal{Z}|^2$ in the logarithm accounting for a union bound over arms). The $(1 + \epsilon)$ accounts for slack from the rounding principle. Finally, this confidence interval allows us to provably remove any arm $z \in \widehat{\mathcal{Z}}_t$ such that $\Delta(z) > 2^{-(t-1)}$ in round $t$.

---

**Algorithm 1**: **RAGE**$(\mathcal{X}, \mathcal{Z}, \epsilon, r(\cdot), \delta)$: **R**andomized **A**daptive **G**ap **E**limination

**Input**: Arms $\mathcal{X} \subset \mathbb{R}^d$, items $\mathcal{Z} \subset \mathbb{R}^d$, rounding approximation factor $\epsilon$ with default value $1/10$, function $r(\cdot)$ giving minimum number of samples to obtain rounding approximation $\epsilon$, and confidence level $\delta \in (0, 1)$.

**Initialize**: Let $\widehat{\mathcal{Z}}_1 \leftarrow \mathcal{Z}, t \leftarrow 1$

**while** $|\widehat{\mathcal{Z}}_t| > 1$ **do**

   $\delta_t \leftarrow \frac{\delta}{t^2}$

   $\lambda_t^* \leftarrow \arg\min_{\lambda \in \triangle_{\mathcal{X}}} \max_{y \in \mathcal{Y}(\widehat{\mathcal{Z}}_t)} \|y\|^2_{(\sum_{x \in \mathcal{X}} \lambda_x x x^\top)^{-1}}$

   $\rho(\mathcal{Y}(\widehat{\mathcal{Z}}_t)) \leftarrow \min_{\lambda \in \triangle_{\mathcal{X}}} \max_{y \in \mathcal{Y}(\widehat{\mathcal{Z}}_t)} \|y\|^2_{(\sum_{x \in \mathcal{X}} \lambda_x x x^\top)^{-1}}$

   $N_t \leftarrow \max\left\{ \lceil 2(2^t)^2 \rho(\mathcal{Y}(\widehat{\mathcal{Z}}_t))(1+\varepsilon)\log(|\mathcal{Z}|^2/\delta_t) \rceil, r(\epsilon) \right\}$

   $\mathbf{x}_{N_t} \leftarrow \text{ROUND}(\lambda_t^*, N_t)$

   Pull arms $x_1, \ldots, x_{N_t}$ and obtain rewards $r_1, \ldots, r_{N_t}$

   Compute $\widehat{\theta}_t = A_t^{-1} b_t$ using $A_t := \sum_{j=1}^{N_t} x_j x_j^\top$ and $b_t := \sum_{j=1}^{N_t} x_j r_j$

   $\widehat{\mathcal{Z}}_{t+1} \leftarrow \widehat{\mathcal{Z}}_t \setminus \left\{ z \in \widehat{\mathcal{Z}} | \exists z' \in \widehat{\mathcal{Z}} : \|z' - z\|_{A_t^{-1}} \sqrt{2\log(|\mathcal{Z}|^2/\delta_t)} < (z' - z)^\top \widehat{\theta}_t \right\}$

   $t \leftarrow t + 1$

**Output**: $\widehat{\mathcal{Z}}_t$

---

**Theorem 2.** *Assume that $\max_{z \in \mathcal{Z}} \Delta(z) \le 2$. Then with probability at least $1 - \delta$, using an $\epsilon$-efficient rounding procedure, Algorithm 1 returns $z_*$ and requires a worst-case sample complexity of*

$$N \le \sum_{t=1}^{\lfloor \log_2(4/\Delta_{\min}) \rfloor} \max\left\{ \lceil 2(2^t)^2 \rho(\mathcal{Y}(\mathcal{S}_t))(1+\epsilon)\log(t^2 |\mathcal{Z}|^2/\delta) \rceil, r(\epsilon) \right\}$$

*where $\mathcal{S}_t = \{z \in \mathcal{Z} : \Delta(z) \le 4 \cdot 2^{-t}\}$. In particular, ROUND can be chosen so that $r(\epsilon) = O(d/\epsilon^2)$. Furthermore, $N \le c\psi^* \log_2(4/\Delta_{\min}) \log(|\mathcal{Z}|^2 \log_2(4/\Delta_{\min})^2/\delta) + r(\epsilon) \log_2(4/\Delta_{\min})$ for some absolute constant $c$, in other words Algorithm 1 is instance optimal up to logarithmic factors.*

We provide a proof of the sample complexity bound in Section A. The primary novelty in our analysis is in quantifying the relationship between the algorithm sample complexity and the lower bound.

## 3.1 Interpreting the sample complexity.

Up to logarithmic factors, Algorithm 1 matches the lower bound obtained in Theorem 1. However, the term $\rho(\mathcal{Y}(\mathcal{S}_t))$ may seem a bit mysterious. In this section we try to interpret this quantity in terms of the geometry of $\mathcal{X}$ and $\mathcal{Z}$.

Let $\text{conv}(\mathcal{X} \cup -\mathcal{X})$ denote the convex hull of $\mathcal{X} \cup -\mathcal{X}$, and for any set $\mathcal{Y} \subset \mathbb{R}^d$ define the gauge of $\mathcal{Y}$,

$$\gamma_{\mathcal{Y}} = \max\{c > 0 : c\mathcal{Y} \subset \text{conv}(\mathcal{X} \cup -\mathcal{X})\}.$$

In the case where $\mathcal{Y}$ is a singleton $\mathcal{Y} = \{y\}$, $\gamma(y) := \gamma_{\mathcal{Y}}$ is the *gauge norm* of $y$ with respect to $\text{conv}(\mathcal{X} \cup -\mathcal{X})$, a familiar quantity from convex analysis [28]. We can provide a natural upper bound for $\rho(\mathcal{Y})$ in terms of the gauge.

**Lemma 1.** *Let $\mathcal{Y} \subset \mathbb{R}^d$. Then*

$$\max_{y \in \mathcal{Y}} \|y\|_2^2 / (\max_{x \in \mathcal{X}} \|x\|_2) \leq \rho(\mathcal{Y}) \leq d/\gamma_{\mathcal{Y}}^2. \tag{4}$$

*In the case of a singleton $\mathcal{Y} = \{y\}$, we can improve the upper bound to $\rho(\mathcal{Y}) \leq 1/\gamma(y)^2$.*

The proof of this Lemma is in Appendix E. To see the potential for adaptive gains we focus on the case of linear bandits where $\mathcal{X} = \mathcal{Z}$. Consider an example with $\mathcal{X} = \{e_i\}_{i=1}^d \cup \{z'\}$ for $z' = (\cos(\alpha), \sin(\alpha), 0, \cdots, 0)$ where $\alpha \in [0, .1)$, and $\theta^* = e_1$. Note that $\Delta_{\min} \approx 1 - \cos(\alpha) \approx \alpha^2/2$. Then $\mathcal{S}_1 = \mathcal{X}$, and an easy computation shows $\gamma_{\mathcal{Y}(\mathcal{X})}$ is a constant bounded from zero. After the first round, all arms except $e_1$ and $z'$ will be removed, so $\mathcal{Y}(\mathcal{S}_t) = \{e_1 - z'\}$ for $t \geq 2$, and $\gamma_{\mathcal{Y}(\mathcal{S}_t)} \approx 1/\sin(\alpha) \approx 1/\alpha$. Summing over all rounds, we see that this implies a sample complexity of $\widetilde{O}(d + 1/\alpha^2)$ up to log factors, which is a significant improvement over the static $\mathcal{X}\mathcal{Y}$-allocation sample complexity of $\widetilde{O}(d/\alpha^2)$.

## 4 Related Work

When $\mathcal{X} = \mathcal{Z} = \{e_1, \cdots, e_d\} \subset \mathbb{R}^d$ is the set of standard basis vectors, the problem reduces to that of the best-arm identification problem for multi-armed bandits which has been extensively studied [14, 19, 20, 22, 11]. In addition, pure exploration for combinatorial bandits where $\mathcal{X} = \{e_1, \cdots, e_d\} \subset \mathbb{R}^d$ and $\mathcal{Z} \subset \{0,1\}^d$ has also received a great deal of attention [10, 8, 12, 9].

In the setting of linear bandits when $\mathcal{X} = \mathcal{Z}$, despite a great deal of work in the regret and contextual settings [1, 26, 25, 13], there has been far less work on linear bandits for pure exploration. This problem was first introduced in [30] and since then, there have been a few other works on this topic, [31, 21, 33] that we now discuss.

- Soare et al. [30] made the initial connections to G-optimal experimental design. That work provides the first passive algorithm with a sample complexity of $O(\frac{d}{\Delta_{\min}^2} \log(|\mathcal{X}|/\delta) + d^2)$. Note that the $d^2$ comes from the minimum number of samples needed for an efficient rounding procedure and thus could be reduced to $d$ using improved rounding procedures (see [2]). They also provide an adaptive algorithm, $\mathcal{X}\mathcal{Y}$-adaptive algorithm for linear bandits. Their algorithm is very similar to ours, with two notable differences. Firstly, instead of using an efficient rounding procedure, they use a greedy iterative scheme to compute an optimal allocation. Secondly, their algorithm does not discard items that are provably sub-optimal. As a result, their sample complexity (up to logarithmic factors) scales as $\max\{M^*, \psi^*\} \log(|\mathcal{X}|/(\Delta_{\min}\delta)) + d^2$ where $M^*$ is defined (informally) as the amount of samples needed using a static allocation to remove all sub-optimal directions in $\mathcal{Y}(\mathcal{X}) \setminus \mathcal{Y}^*(\mathcal{X})$.

- In Tao et al. [31], the focus is on developing different estimators with the goal of removing the constant term $d^2$ in Soare et al.'s passive sample complexity. Instead of using a rounding procedure, they use a different estimator than the OLS estimator $\theta^*$. Note that the rounding procedure in [2] and described in the supplementary could have been applied directly to Soare's static allocation algorithm giving the same sample complexity as the one obtained in [31]. They also provide an adaptive algorithm *ALBA*, that achieves a sample complexity of $O(\sum_{i=1}^d 1/\Delta_i^2)$ where $\Delta_i$ is the $i$-th smallest gap of the vectors in $\mathcal{X}$. It is easy to see that this sample complexity is not optimal: imagine a situation in which the vectors of $\mathcal{X}$ with the $(d-1)$-smallest gaps are identical to the vector $x' \neq x^*$. Then we only need to pay once for the samples needed to remove $x'$, not

$(d-1)$-times. Finally, their algorithms do not compute the optimal allocation over differences of vectors in $\mathcal{X}$, but instead on $\mathcal{X}$ directly à la G-optimal design. We will see the inefficiency of this strategy in the experiments.

- Karnin [21] provides an algorithm that uses repeated rounds (for probability amplification) of exploration phases combined with verification phases to provide an asymptotically optimal algorithm, meaning when $\delta \to 0$ the sample complexity divided by $\log(1/\delta)$ approaches $\psi^*$. Though this is a nice theoretical result, the algorithm is not practical; the exploration phase is simply a naïve passive $G$-optimal design.

- In Xu et al. [33], a fully adaptive algorithm called LinGapE inspired by the UGapE algorithm [15] is proposed. Since LinGapE is fully adaptive, a confidence bound allowing for dependence in the samples is necessary and the authors employ the self-normalized bound of [1]. The algorithm requires each arm to be pulled once - an undesirable characteristic of a linear bandit algorithm since the structure of the problem allows for information to be obtained about arms that are not pulled. A recent work [23], extends this algorithm to generalized linear models where the expected reward of pulling arm $z$ reward is given by a non-linear link function of $z^\top \theta^*$.

Finally, we mention [34], which considers transductive experimental design from a computational and optimization perspective, and explores $\mathcal{X}\mathcal{Y}$-allocation for arbitrary kernels.

# 5 Experiments

In this section, we present simulations for the linear bandit pure exploration problem and the general transductive bandit problem. We compare our proposed algorithm with both adaptive and non-adaptive strategies. The adaptive strategies are $\mathcal{X}\mathcal{Y}$-Adaptive allocation from [30], LinGapE from [33], and ALBA from [31]. The non-adaptive strategies are static $\mathcal{X}\mathcal{Y}$-allocation, as described in Section 2, and an oracle strategy that knows $\theta^*$ and samples according to $\lambda^*$. We do not compare to the algorithm given in [21] since it is primarily a theoretical contribution and in moderate-confidence regimes obtains only the non-adaptive sample complexity. We run each algorithm at a confidence level of $\delta = 0.05$. The empirical failure probability of each of the algorithms in the simulations is zero. To compute the samples for RAGE, we first used the Frank-Wolfe algorithm (with a precise stopping condition in the supplementary) to find $\lambda_t$, and then the rounding procedure from [27] with $\epsilon = 1/10$. Further implementation details of RAGE and discussion pertaining to the implementation of the other algorithms can be found in the supplementary material in Section F. We remark here that in our implementation of the $\mathcal{X}\mathcal{Y}$-Adaptive allocation, we follow the experiments in [30] and allow for provably suboptimal arms to be discarded (though this is not how the algorithm is written in their paper). The resulting algorithm is then similar to our algorithm. Unless explicitly mentioned, noise in the observations was generated from a standard normal distribution.

**Linear bandits: benchmark example.** The first experiment we present has become a benchmark in the linear bandit pure exploration literature since it was introduced in [30]. In this problem, $\mathcal{X} = \mathcal{Z} = \{e_1, \ldots, e_d, x'\} \subset \mathbb{R}^d$ where $e_i$ is the $i$-th standard basis vector, $x' = \cos(.01)e_1 + \sin(.01)e_2$, and $\theta^* = 2e_1$ so that $x_* = x_1$. An efficient sampling strategy for this problem needs to focus on reducing uncertainty in the direction $(x_1 - x_{d+1})$, which can be achieved by focusing pulls on arm $x_2 = e_2$ since it is most aligned with this direction.

The results for this experiment are shown in Fig. 1a. The RAGE algorithm performs competitively with existing algorithms and the oracle allocation. The $\mathcal{X}\mathcal{Y}$-Adaptive algorithm is similar to RAGE, but with weaker theoretical guarantees, so naturally it performs nearly equivalently. We omit it from the remaining experiments for this reason. The LinGapE algorithm performs well when the number of dimensions and arms is small. However, as the number of arms grows, LinGapE suffers from a worse dimension dependency in the confidence interval. ALBA performs the worst of the recently proposed algorithms and this is to be expected since it computes an allocation on the $\mathcal{X}$ set instead of on the $\mathcal{Y}(\mathcal{X})$ set. This example clearly highlights the gains of adaptive sampling over non-adaptive allocations such as the static $\mathcal{X}\mathcal{Y}$-allocation. However, since $\mathcal{X}$ is relatively small in this case, it fails to tease out important differences between the algorithms that can greatly increase the sample complexity. We construct examples to demonstrate these claims now.

**Many arms with moderate gaps.** In this example, for a given value of $n \geq 3$, we construct a set of arms $\mathcal{X} \subset \mathbb{R}^2$, where $\mathcal{X} = \mathcal{Z} = \{e_1, \cos(3\pi/4)e_1 + \sin(3\pi/4)e_2\} \cup \{\cos(\pi/4 + \phi_i)e_1 + \sin(\pi/4 + \phi_i)e_2\}_{i=3}^n$ with $\phi_i \sim \mathcal{N}(0, .09)$ for each $i \in \{3, \ldots, n\}$. The parameter vector is fixed to be $\theta^* = e_1$

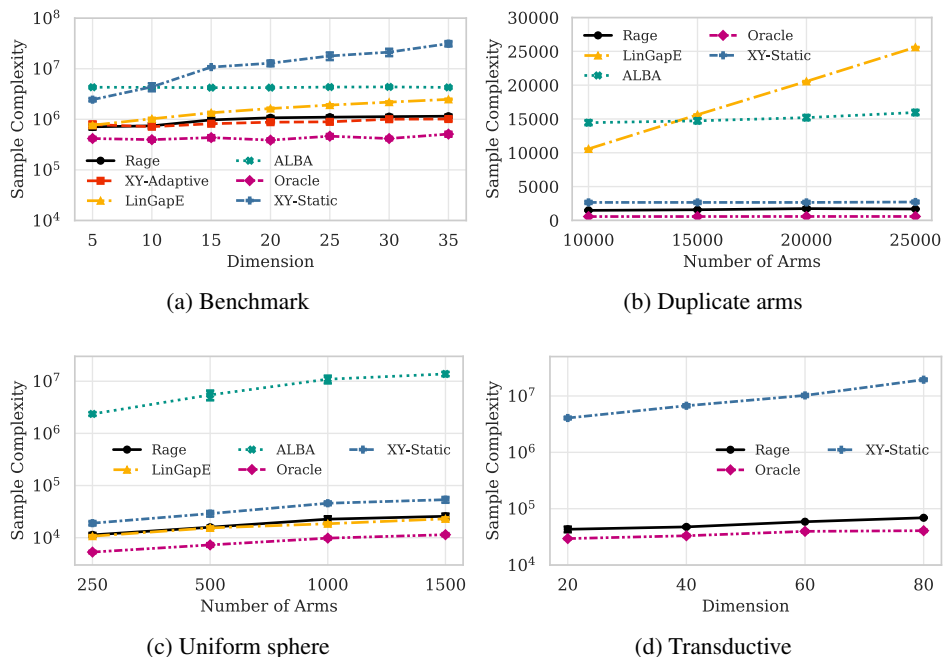

| (a) Benchmark | (b) Duplicate arms |
| --- | --- |
| (c) Uniform sphere | (d) Transductive |

Figure 1

so that $x_1$ is the optimal arm, $x_2$ gives the most information to identify the optimal arm, and the remaining arms roughly point in the same direction with an expected gap of $\Delta \approx 0.3$.

In Fig. 1b, we show the results of the experiment as we increase the number of arms. The LinGapE algorithm suffers from a linear scaling in the number of arms since it must sample each arm as an initialization. An efficient sampling strategy should focus energy on $x_2$, and as it does so, it will gain information about the arms that are nearly duplicates of each other, which is how RAGE performs.

**Uniform distribution on a sphere.** In this example, $\mathcal{X} = \mathcal{Z}$ is sampled from a unit sphere of dimension $d = 9$ centered at the origin. Following [31], we select the two closest arms $x, x' \in \mathcal{X}$ and let $\theta^* = x$. In Fig. 1c, we show the sample complexity of the algorithms as the number of arms grows. The RAGE algorithm significantly outperforms ALBA and this is primarily due to the fact that ALBA computes a G-optimal design on the active vectors in each round instead of on the differences between these vectors. Thus the ALBA sampling distribution can be focused on a very different set of arms from the optimal one.

**Transductive example.** We now present a general transductive bandit example. Since the existing algorithms in the linear bandit literature do not generalize to this problem, we compare with a static $\mathcal{X}\mathcal{Y}$-allocation on $\mathcal{X}, \mathcal{Y}(\mathcal{Z})$ and an oracle $\mathcal{X}\mathcal{Y}$-allocation on $\mathcal{X}, \mathcal{Y}^*(\mathcal{Z})$ that knows the optimal arm and the gaps. We construct an example in $\mathbb{R}^d$ with $d$ even where $\mathcal{X} = \{e_1, \ldots, e_d\}$. The set $\mathcal{Z}$ is also chosen so $|\mathcal{Z}| = d$, the first $d/2$ vectors are given by $z_1, \ldots, z_{d/2} = (e_1, \ldots, e_{d/2})$ and then $z_{d/2+j} = \cos(.1)e_j + \sin(.1)e_{j+d/2}$ for each $j \in \{1, \ldots, d/2\}$. Take $\theta^* = e_1$ so $z_1$ is the optimal arm. The results of this simulation are depicted in Fig. 1d. The RAGE algorithm significantly outperforms the static allocation and nearly matches the oracle allocation.

We now present examples motivated by real-world applications.

**Multivariate testing example.** In many experimental design settings, there are a series of $D$ factors that can be either in a set of $N$ states, and the goal is to determine the treatment configuration that has the highest outcome for a given metric. As a concrete example in web page optimization, it is common that the composition of an advertisement layout selection may consist of several choices such as an image, background color, and keyword to display (e.g. [16]), and we seek to find the combination with the highest clickthrough rate. To formalize the problem, consider a webpage consisting of $D$ distinct slots and suppose that there are 2 content choices that can be presented in each slot. Let

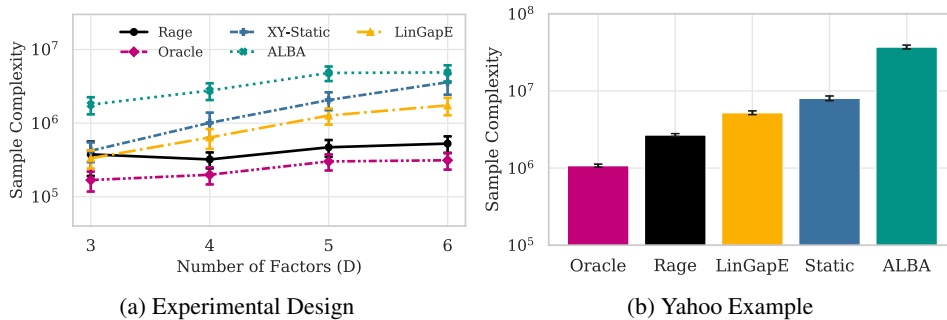

(a) Experimental Design          (b) Yahoo Example

Figure 2

the set $\mathcal{W} = \{-1, 1\}^D$ satisfying $|\mathcal{W}| = 2^D$ encode each layout. We model the problem using a factorial design (see, e.g., [6]) including pairwise interaction features to generate a linear bandit problem. Each layout is represented by an arm $x \in \mathcal{X}$ where $\mathcal{X} = \mathcal{Z} \subset \{-1, 1\}^{1+D+D(D-1)/2}$ and $|\mathcal{X}| = 2^D$. The expected reward of any $x \in \mathcal{X}$ corresponding to a layout $w \in \mathcal{W}$ is given by

$$x^\top \theta^* = \theta_0^* + \alpha_1 \sum_{j=1}^{D} \theta_j^* w_j + \alpha_2 \sum_{k=1}^{D} \sum_{\ell=k+1}^{D} \theta_{k,\ell}^* w_k w_\ell,$$

where $\theta_0^*$ is a common bias weight, $\theta_j^*$ is a weight for the $j$–th slot, and $\theta_{j,k}^*$ is a weight for the interaction between the content in the $k$–th and $\ell$–th slots. We also include known parameters $\alpha_1 = 1$ and $\alpha_2 = 0.5$ that control the strength of the first and second order interactions respectively. The weights of the parameter vector are drawn from a discrete uniform distribution with a range of $[-0.3, 0.3]$ and a granularity of $0.01$. The results of this example are shown in Fig. 2a. The RAGE algorithm performs close to the oracle on this example, while the sample complexity of the rest of the algorithms grows as the number of arms and dimension of the problem goes up.

**Click-through example.** To conduct an experiment based on real data, we build a problem using the Yahoo! Webscope Dataset R6A.[5] The dataset contains user click log records for news articles displayed uniformly at random on the Yahoo! front page between May 1st, 2009 and May 10th, 2009. Each click log record consists of a binary outcome along with 6 features identifying the user and 6 features identifying the article.

To build a linear bandit problem from the dataset, we construct an arm set $\mathcal{X} = \mathcal{Z} \subset \mathbb{R}^{36}$ by taking the outer product of the user and article features for each click log record on May 1st, 2009. We then fit a regularized least squares estimate using a regularization parameter of $0.01$ to obtain $\theta^*$. To model binary rewards, we let the observed reward be generated by a draw of a Bernoulli random variable with parameter $x^\top \theta^*$ for any arm selection $x \in \mathcal{X}$. Since $x^\top \theta^* \in (0, 0.11) \ \forall \ x \in \mathcal{X}$, the noise is bounded between $[-1, 1]$, which causes it to be 1-sub-Gaussian. We simulate the problem with 40 arms including the arm with the maximum reward in the dataset and the remaining arms were selected at random from the set of arms with gap at least $0.01$ from the optimal arm so the problem is not too hard. The experiment setup is similar to that from [33] for this dataset. The results are presented in Fig. 2b. We see that the RAGE algorithm has good performance on this example based on real world data.

## 6 Conclusion

In this paper we have proposed the problem of best-arm identification for transductive linear bandits, provided an algorithm, and matching upper and lower bounds. As a remark it is straightforward to exit our algorithm early with an $\varepsilon$-good arm. It still remains to develop anytime algorithms for this problem, as has been done in pure exploration for multi-armed bandits [19] that do not throw out samples. In addition, we suspect our algorithm actually matches the lower-bound and the $\log(1/\Delta_{\min})$ factor is unnecessary. Finally, it is possible that some of the ideas developed here extend to the setting of regret and could be used to give instance based regret bounds for linear bandits [25]. We hope to explore connections to both the regret and fixed budget settings in further works.

## Footnotes

[2]Such an assumption is avoided by a sophisticated rounding procedure that we will describe shortly.

[3]There is a technical issue of whether the set $\mathcal{Z}$ lies in the span of $\mathcal{X}$ which in general is necessary to obtain unbiased estimates of $(z_* - z)^\top \theta^*$. Throughout the following we assume that $\text{span}(\mathcal{X}) = \mathbb{R}^d$.

[4] Where we recall for any subset $\mathcal{S} \subset \mathcal{Z}$, $\mathcal{Y}(\mathcal{S}) := \{z - z' : z, z' \in \mathcal{S}\}$ and for an arbitrary subset $\mathcal{V} \subset \mathbb{R}^d$ we have $\rho(\mathcal{V}) = \min_{\lambda \in \triangle_{\mathcal{X}}} \max_{v \in \mathcal{V}} \|v\|^2_{(\sum_{x \in \mathcal{X}} \lambda_x x x^\top)^{-1}}$.

[5] https://webscope.sandbox.yahoo.com/

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
