[Supplementary Material]

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

# A Proof of Theorem 2

*Proof.* Let the good event for the $t$–th round of Algorithm 1 be

$$\mathcal{E}_t := \left\{ N_t \leq \max\{\lceil 2(2^t)^2 \rho(\mathcal{Y}(\mathcal{S}_t))(1+\epsilon) \log\left(\frac{|\mathcal{Z}|^2}{\delta_t}\right)\rceil, r(\epsilon)\}\right\} \cap \{z_* \in \widehat{\mathcal{Z}}_{t+1}\} \cap \{\widehat{\mathcal{Z}}_{t+1} \subseteq \mathcal{S}_{t+1}\}$$

where we recall that $\mathcal{S}_t = \{z \in \mathcal{Z} : \Delta(z) \leq 4 \cdot 2^{-t}\}$. The good event characterizes the worst-case sample complexity of the $t$–th phase of Algorithm 1 and guarantees that the set of active arms at the end of the phase contains the optimal arm and it is contained in the set of arms with gaps below the threshold to be eliminated in the phase. Note that for $t > \log_2(4/\Delta_{\min})$ we have $\mathcal{S}_t = \{z_*\}$.

The proof proceeds as follows. We begin by showing that the good event holds with probability at least $1 - \delta_t$ in phase $t$ given that the good event held in phase $t-1$. We then show that the probability of the good event holding in every phase is at least $1 - \delta$. As a result, we simply sum over the bound on the sample complexity in each phase given in the good event to obtain the stated bound on the sample complexity.

The following lemma shows that good event holds in phase $t$ with probability at least $1-\delta_t$ conditioned on the good event holding in the previous phases.

**Lemma 2.** $\mathbb{P}(\mathcal{E}_t | \mathcal{E}_{t-1}, \cdots, \mathcal{E}_1) \geq 1 - \delta_t.$

*Proof.* Conditioned on a choice of $\mathcal{Y}(\widehat{\mathcal{Z}}_t)$, since $\widehat{\theta}$ is a least squares estimator of $\theta^*$ and the noise is i.i.d., we know that $y^\top(\theta^* - \widehat{\theta}_t)$ is $\|y\|^2_{A_t^{-1}}$-subGaussian for all $y \in \mathcal{Y}(\widehat{\mathcal{Z}}_t)$. Furthermore, due to the guarantees of the rounding procedure, $\|y\|^2_{A_t^{-1}} \leq (1+\epsilon)\rho(\mathcal{Y}(\widehat{\mathcal{Z}}_t))/N_t \leq \left(2(2^t)^2 \log(|\mathcal{Z}|^2/\delta_t)\right)^{-1}$ for all $y \in \mathcal{Y}(\widehat{\mathcal{Z}}_t)$ by our choice of $N_t$. Since the right-hand side is deterministic, independent of $\mathcal{Y}(\widehat{\mathcal{Z}}_t)$, for any $\nu > 0$, we have that

$$\mathbb{P}\left( |y^\top(\theta^* - \widehat{\theta})| > \sqrt{\frac{2\log(2/\nu)}{2(2^t)^2 \log(|\mathcal{Z}|^2/\delta_t)}} \middle| \mathcal{E}_{t-1}, \cdots, \mathcal{E}_1 \right) \leq \nu$$

for any $y \in \mathcal{Y}(\widehat{\mathcal{Z}}_t)$. Taking $\nu = 2\delta_t/|\mathcal{Z}|^2$ and union bounding over all the possible $y \in \mathcal{Y}(\widehat{\mathcal{Z}}_t)$ where $|\mathcal{Y}(\widehat{\mathcal{Z}}_t)| \leq |\mathcal{Y}(\mathcal{Z})| \leq |\mathcal{Z}|^2/2$, gives us that

$$\mathbb{P}(\exists y \in \mathcal{Y}(\widehat{\mathcal{Z}}_t) \quad |y^\top(\theta^* - \widehat{\theta})| > 2^{-t} | \mathcal{E}_{t-1}, \cdots, \mathcal{E}_1) \leq \delta_t. \tag{5}$$

*Claim 1:* Every arm $z \in \widehat{\mathcal{Z}}_t$ such that $\Delta(z) > 2^{-(t-1)}$ is discarded in phase $t$ so that $\widehat{\mathcal{Z}}_{t+1} \subseteq \mathcal{S}_{t+1}$ with probability at least $1 - \delta_t$.

**Proof.** Since we conditioned on $\mathcal{E}_{t-1}$, $z_* \in \widehat{\mathcal{Z}}_t$. If $z \in \mathcal{S}_{t+1}^c \cap \widehat{\mathcal{Z}}_t$ then by definition $\Delta(z) = (z_* - z)^\top\theta^* > 2^{-(t-1)}$. Taking $y = z_* - z$, we know that a) $y^\top\theta^* > 2^{-(t-1)}$, and b) from the confidence bound $\|y\|_{A_t^{-1}}\sqrt{2\log(|\mathcal{Z}|^2/\delta_t)} \leq 2^{-t}$. Hence,

$$\begin{aligned}
y^\top\widehat{\theta}_t &\geq y^\top\theta^* - \|y\|_{A_t^{-1}}\sqrt{2\log(|\mathcal{Z}|^2/\delta_t)} \\
&\overset{a)}{>} 2^{-(t-1)} - \|y\|_{A_t^{-1}}\sqrt{2\log(|\mathcal{Z}|^2/\delta_t)} \\
&\overset{b)}{\geq} 2^{-(t-1)} - 2^{-t} \\
&= 2^{-t} \overset{b)}{\geq} \|y\|_{A_t^{-1}}\sqrt{2\log(|\mathcal{Z}|^2/\delta_t)}
\end{aligned}$$

However, this is precisely the discard condition of the algorithm guaranteeing $z$ will be eliminated.

We now show that the optimal arm cannot be discarded in a phase with high probability. $\qquad\square$

*Claim 2:* $z_* \in \widehat{\mathcal{Z}}_{t+1}$ with probability at least $1 - \delta_t$.

**Proof.** We prove this claim by contradiction. To begin, observe that $z_*$ is in $\widehat{\mathcal{Z}}_t$ since $\mathcal{E}_{t-1}$ holds. Now, suppose that $z_*$ is discarded in phase $t$. This implies that there exists a $z \neq z_*$ for $z \in \widehat{\mathcal{Z}}_t$

such that $\|z - z^*\|_{A_t^{-1}} \sqrt{2\log(|\mathcal{Z}^2|/\delta_t)} < (z - z_*)^\top \widehat{\theta}_t$. However from the confidence interval (5), $(z-z_*)^\top(\widehat{\theta}_t - \theta^*) \leq \|z - z^*\|_{A_t^{-1}} \sqrt{2\log(|\mathcal{Z}^2|/\delta_t)}$. Combining these we see that $(z-z_*)^\top(\widehat{\theta}_t - \theta^*) < (z - z_*)^\top \widehat{\theta}_t$ which implies $(z - z_*)^\top \theta^* > 0$ which is a contradiction. $\qquad \square$

We complete the proof by showing that the sample complexity of phase $t$ given in the good event holds with probability $1 - \delta_t$. Since $\mathcal{E}_{t-1}$ is given, $\widehat{\mathcal{Z}}_t \subseteq \mathcal{S}_t$, which implies with probability at least $1 - \delta_t$,

$$N_t = \max\left\{\left\lceil 2(2^t)^2 \rho(\mathcal{Y}(\widehat{\mathcal{Z}}_t))(1 + \varepsilon)\log(|\mathcal{Z}|^2/\delta_t)\right\rceil, r(\epsilon)\right\}$$
$$\leq \max\left\{\left\lceil 2(2^t)^2 \rho(\mathcal{Y}(\mathcal{S}_t))(1 + \varepsilon)\log(|\mathcal{Z}|^2/\delta_t)\right\rceil, r(\epsilon)\right\}$$

where we note that the quantity on the right hand side is deterministic. $\qquad \square$

**Lemma 3.** $\mathbb{P}(\mathcal{E}_1 \cap \cdots \cap \mathcal{E}_{\lfloor \log_2(4/\Delta_{\min})\rfloor}) \geq 1 - \delta$.

*Proof.* Let us first expand the intersection of the events into a product of conditional probabilities as follows:
$$\mathbb{P}(\mathcal{E}_1 \cap \cdots \cap \mathcal{E}_{\lfloor \log_2(4/\Delta_{\min})\rfloor}) = \Pi_{t=1}^{\lfloor \log_2(4/\Delta_{\min})\rfloor} \mathbb{P}(\mathcal{E}_t | \mathcal{E}_{t-1} \cap \cdots \cap \mathcal{E}_1)$$
We now obtain a lower bound on the success probability using Lemma 2 and facts about infinite products:

$$\Pi_{t=1}^{\lfloor \log_2(4/\Delta_{\min})\rfloor} \mathbb{P}(\mathcal{E}_t | \mathcal{E}_{t-1} \cap \cdots \cap \mathcal{E}_1) \geq \Pi_{t=1}^{\lfloor \log_2(4/\Delta_{\min})\rfloor}(1 - \delta_t) \geq \Pi_{t=1}^{\infty}\left(1 - \frac{\delta}{t^2}\right) = \frac{\sin(\pi\delta)}{\pi\delta}.$$

Finally, using the fact that $\frac{\sin(\pi\delta)}{\pi\delta} \geq 1 - \delta$ for $\delta \in (0,1)$, we obtain the result $\mathbb{P}(\mathcal{E}_1 \cap \cdots \cap \mathcal{E}_{\lfloor \log_2(4/\Delta_{\min})\rfloor}) \geq 1 - \delta$. $\qquad \square$

The sample complexity follows immediately from Lemmas 2 and 3 since we can now sum the number of samples taken in each phase. However, a key novelty of this proof is the the following method to quantify the relationship between the algorithm sample complexity and the lower bound. With probability at least $1 - \delta$,

$$N \leq \sum_{t=1}^{\lfloor \log_2(4/\Delta_{\min})\rfloor} \max\left\{\left\lceil 2(2^t)^2 \rho(\mathcal{Y}(\mathcal{S}_t))(1 + \epsilon)\log(t^2|\mathcal{Z}|^2/\delta)\right\rceil, r(\epsilon)\right\}$$
$$\leq 128\psi^*(1 + \epsilon)\log_2(4/\Delta_{\min})\log(\log_2(4/\Delta_{\min})^2|\mathcal{Z}|^2/\delta) + (1 + r(\epsilon))\log_2(4/\Delta_{\min}).$$

Recall that $\mathcal{Y}^*(\mathcal{S}) = \{z_* - z : \forall z \in \mathcal{S} \setminus z_*\}$. To see the second inequality, note that

$$\psi^* = \min_{\lambda \in \triangle_\mathcal{X}} \max_{y \in \mathcal{Y}^*(\mathcal{Z})} \frac{\|y\|_{(\sum_{x \in \mathcal{X}} \lambda_x xx^\top)^{-1}}^2}{\Delta(y)^2}$$

$$= \min_{\lambda \in \triangle_\mathcal{X}} \max_{t \leq \lfloor \log_2(4/\Delta_{\min})\rfloor} \max_{y \in \mathcal{Y}^*(\mathcal{S}_t)} \frac{\|y\|_{(\sum_{x \in \mathcal{X}} \lambda_x xx^\top)^{-1}}^2}{\Delta(y)^2}$$

$$\geq \min_{\lambda \in \triangle_\mathcal{X}} \max_{t \leq \lfloor \log_2(4/\Delta_{\min})\rfloor} \max_{y \in \mathcal{Y}^*(\mathcal{S}_t)} \frac{\|y\|_{(\sum_{x \in \mathcal{X}} \lambda_x xx^\top)^{-1}}^2}{(4 \cdot 2^{-t})^2}$$

$$\overset{(i)}{\geq} \frac{1}{16\log_2(1/\Delta_{\min})} \min_{\lambda \in \triangle_\mathcal{X}} \sum_{t=1}^{\lfloor \log_2(4/\Delta_{\min})\rfloor} (2^t)^2 \max_{y \in \mathcal{Y}^*(\mathcal{S}_t)} \|y\|_{(\sum_{x \in \mathcal{X}} \lambda_x xx^\top)^{-1}}^2$$

$$\overset{(ii)}{\geq} \frac{1}{16\log_2(4/\Delta_{\min})} \sum_{t=1}^{\lfloor \log_2(4/\Delta_{\min})\rfloor} (2^t)^2 \min_{\lambda \in \triangle_\mathcal{X}} \max_{y \in \mathcal{Y}^*(\mathcal{S}_t)} \|y\|_{(\sum_{x \in \mathcal{X}} \lambda_x xx^\top)^{-1}}^2$$

$$\overset{(iii)}{\geq} \frac{1}{64\log_2(4/\Delta_{\min})} \sum_{t=1}^{\lfloor \log_2(4/\Delta_{\min})\rfloor} (2^t)^2 \min_{\lambda \in \triangle_\mathcal{X}} \max_{y \in \mathcal{Y}(\mathcal{S}_t)} \|y\|_{(\sum_{x \in \mathcal{X}} \lambda_x xx^\top)^{-1}}^2$$

$$= \frac{1}{64\log_2(4/\Delta_{\min})} \sum_{i=1}^{\lfloor \log_2(4/\Delta_{\min})\rfloor} (2^t)^2 \rho(\mathcal{Y}(\mathcal{S}_t))$$

where $(i)$ follows from the fact that the maximum of positive numbers is always greater than the average, and $(ii)$ by the fact that the minimum of a sum is greater than the sum of minimums. To see $(iii)$, note that for $y \in \mathcal{Y}(\mathcal{S}_t)$, if $y = z_i - z_j$, then $y = (z_* - z_j) - (z_* - z_i)$. Hence $\max_{y \in \mathcal{Y}(\mathcal{S}_t)} \|y\|^2_{(\sum_{x \in \mathcal{X}} \lambda_x x x^\top)^{-1}} \le 4 \max_{y \in \mathcal{Y}^*(\mathcal{S}_t)} \|y\|^2_{(\sum_{x \in \mathcal{X}} \lambda_x x x^\top)^{-1}}$. $\qquad \square$

# B  Efficient Rounding Procedures

Throughout the following we assume that $\mathcal{Y} \subset \mathbb{R}^d$ is arbitrary and that $\mathcal{X} = \{x_1, \cdots, x_n\} \subset \mathbb{R}^d$ is a subset with $\dim \operatorname{span}(\mathcal{X}) = d$.

**Definition 2.** *A rounding procedure is an algorithm that takes as input $\lambda \in \triangle^n$, a set of vectors $\mathcal{X}$, and a number of samples $N$ and returns a finite **allocation** $s = (s_1, \cdots, s_n) \in \mathbb{N}^n$ satisfying the following properties: 1. $\sum_{i=1}^n s_i = N$; 2. there exists a function $r(\epsilon)$ such that if $N > r(\epsilon)$, then $\max_{y \in \mathcal{Y}} \|y\|^2_{(\sum_{i=1}^n s_i x_i x_i^\top)^{-1}} \le (1 + \epsilon) \max_{y \in \mathcal{Y}} \|y\|^2_{(\sum_{i=1}^n \lambda_i x_i x_i^\top)^{-1}} / N$.*

Fortunately, there has been extensive work on efficient rounding procedures, motivated by the strong connection to G-optimal design in optimal linear experimental design [27]. Here we discuss two important rounding procedures. The first is due to [27] and has an $r(\epsilon) = 2p/\epsilon \le (d(d+1) + 2)/\epsilon$ where $p$ is the support size of $\lambda$.

**Rounding Procedure of [27].** An efficient rounding procedure is given in Chapter 12 of [27] to transform a design $\lambda \in \triangle^n$ into a discrete allocation $s \in \mathbb{N}^n$ for any fixed number of samples $N$. The rounding procedure determines the number of pulls $N_i$ to allocate to each arm $x_i$ in the support of $\lambda$ such that $\sum_{i \le p} N_i = N$ where $p$ is the cardinality of the support of $\lambda$. The discrete allocation from the rounding procedure is obtained in two phases:

1. Given the number of samples $N$ to obtain and the cardinality of the support of $\lambda$, samples to allocate to arms in the support of $\lambda$ are computed using $N_i = \lceil (N - \frac{1}{2}p)\lambda_i \rceil$, where $N_1, N_2, \ldots, N_p$ are positive integers constrained such that $\sum_{i \le p} N_i \ge N$.

2. Following the previous phase of the rounding procedure, loop until the discrepancy $(\sum_{i \le p} N_i) - N = 0$, from either increasing a sample count $N_j$ which obtains $N_j / \lambda_j = \min_{i \le p} N_i / \lambda_i$ to $N_j + 1$, or decreasing a sample count $N_j$ which obtains $(N_j - 1)/\lambda_j = \max_{i \le p}(N_i - 1)/\lambda_i$ to $N_j - 1$.

The efficient design apportionment theorem in Section 12.5 of [27] provides the foundation the procedure. We now provide some details on the efficiency of the procedure.

Let $\gamma = s/N$ represent the fractional allocation corresponding to a finite allocation $s$ satisfying the properties in Definition 2 and obtained from applying the efficient rounding procedure to the distribution $\lambda$. Moreover, define $\upsilon(\mathcal{Y}) := \max_{y \in \mathcal{Y}} \|y\|^2_{(\sum_{x \in \mathcal{X}} \gamma_x x x^\top)^{-1}}$.

**Proposition 2.** *The efficient rounding procedure of [27] guarantees for $N \ge 2p$,*

$$\rho(\mathcal{Y}) \le \upsilon(\mathcal{Y}) \le \left(1 + \frac{2p}{N}\right)\rho(\mathcal{Y}).$$

*Moreover, when $dim(span(\mathcal{X})) = d$ and $N \ge d^2 + d + 2$,*

$$\rho(\mathcal{Y}) \le \upsilon(\mathcal{Y}) \le \left(1 + \frac{d^2 + d + 2}{N}\right)\rho(\mathcal{Y}).$$

*Proof.* Define the minimum likelihood ratio of $\gamma$ relative to $\lambda$ as

$$\zeta(\gamma, \lambda) = \min_{x \in \operatorname{supp}(\lambda)} \frac{\gamma_x}{\lambda_x} = \max\{\kappa \ge 0 : \gamma_x \ge \kappa \lambda_x \text{ for all } x \in \mathcal{X}\}.$$

Observe that $\zeta(\gamma, \lambda) \in [0, 1]$ by definition. As an immediate consequence of this definition, we obtain

$$\sum_{x \in \mathcal{X}} \gamma_x x x^\top \ge \sum_{x \in \mathcal{X}} \zeta(\gamma, \lambda_x) \lambda_x x x^\top \iff \left(\sum_{x \in \mathcal{X}} \gamma_x x x^\top\right)^{-1} \le \frac{1}{\zeta(\gamma, \lambda)} \left(\sum_{x \in \mathcal{X}} \lambda_x x x^\top\right)^{-1}.$$

It follows that for all $y \in \mathcal{Y}$,

$$y^\top \left(\sum_{x \in \mathcal{X}} \gamma_x x x^\top\right)^{-1} y \le \frac{1}{\zeta(\gamma, \lambda)} y^\top \left(\sum_{x \in \mathcal{X}} \lambda_x x x^\top\right)^{-1} y,$$

which implies $\upsilon(\mathcal{Y}) \leq \rho(\mathcal{Y})/\zeta(\gamma, \lambda)$. Since the design $\gamma$ is obtained from an efficient design apportionment, Theorem 12.7 of [27] indicates it obtains the best efficiency bound among all standardized designs of sample size $N$. As a result, Lemma 12.8 from [27] holds and guarantees that for $N \geq 2p$,

$$\frac{1}{\zeta(\gamma, \lambda)} \leq \frac{1}{1 - \frac{p}{N}} \leq 1 + \frac{2p}{N}.$$

Hence, for $N \geq 2p$

$$\rho(\mathcal{Y}) \leq \upsilon(\mathcal{Y}) \leq \left(1 + \frac{2p}{N}\right)\rho(\mathcal{Y}).$$

When $\dim(\text{span}(\mathcal{X})) = d$, Caratheodory's theorem indicates that $p \leq d(d+1)/2 + 1$. Consequently, when $N \geq d^2 + d + 2$, we get

$$\rho(\mathcal{Y}) \leq \upsilon(\mathcal{Y}) \leq \left(1 + \frac{d^2 + d + 2}{N}\right)\rho(\mathcal{Y}).$$

$\square$

**Rounding Procedure of [2].** We refer the reader to Algorithm 1 in [2] for details about their rounding procedure. Here we describe their result and how to modify it to our setting. Let $\mathcal{S}_{b,k} = \{s \in [b]^n : \sum_{i=1}^n s_i \leq k\}$ and a continuous relaxation $\mathcal{C}_{b,k} = \{s \in [0,b]^n : \sum_{i=1}^n s_i \leq k\}$.

**Theorem 3** (Theorem 2.1 of [2]). *Suppose $\epsilon \in (0, 1/3)$, $n \geq k \geq 180d/\epsilon^2$, $b \in [k]$. Let $\pi \in C_{b,k}$, then in polynomial-time (in $n$ and $d$) we can round $\pi$ to an integral solution $\widehat{s} \in S_{b,k}$ satisfying* $\max_{y \in \mathcal{Y}} \|y\|^2_{(\sum \widehat{s}_i x_i x_i^\top)^{-1}} \leq (1 + \epsilon) \max_{y \in \mathcal{Y}} \|y\|^2_{(\sum \pi_i x_i x_i^\top)^{-1}}$.

To apply this theorem to obtain an efficient rounding procedure, consider the following. Given a $\lambda \in \triangle_\mathcal{X}$, and a number of samples $N$, let $\pi = N\lambda$ and consider the case where $b = k = N$. Then $k\lambda \in C_{k,k}$. In general the theorem does not allow $N = k > n$, but we can circumvent this by just duplicating each vector in $\mathcal{X}$ exactly $N$ times. Then the allocation $\widehat{s}$ obtained will satisfy the conditions of the above with $r(\epsilon) = 180d/\epsilon^2$. The authors remark that it is most likely true that $r(\epsilon) = d/\epsilon^2$ suffices, but we are not aware of any such result in the literature.

## C  Proof of Theorem 1

*Proof.* In this section we assume $\mathcal{X} = \{x_1, \cdots, x_n\}$ and $\mathcal{Z} = \{z_1, \cdots, z_m\}$. Without loss of generality, we assume that $z_1 = \text{argmax}_{z_i \in \mathcal{Z}} z_i^\top \theta^*$. Let $\mathcal{C} := \{\theta \in \mathbb{R}^d : \exists i \text{ s.t. } \theta^\top(z_1 - z_i) \leq 0\}$, i.e. $\theta \in \mathcal{C}$ if and only if $z_1$ is not the best arm in the linear bandit instance $(\mathcal{X}, \mathcal{Z}, \theta)$.

We now recall the transportation lemma of [22]. Under a $\delta$-PAC strategy for finding the best arm for the bandit instance $(\mathcal{X}, \mathcal{Z}, \theta^*)$, let $T_i$ denote the random variable which is the number of times arm $i$ is pulled. In addition let $\nu_{\theta,i}$ denote the reward distribution of the $i$-th arm of $\mathcal{X}$, i.e. $\nu_{\theta,i} = \mathcal{N}(x_i^\top \theta, 1)$. Then for any $\theta \in \mathcal{C}$ we have that

$$\sum_{i=1}^n \mathbb{E}[T_i] KL(\nu_{\theta^*,i}, \nu_{\theta,i}) \geq \log(1/2.4\delta).$$

In particular, $\sum_{i=1}^n \mathbb{E}[T_i] \geq \sum_{i=1}^n t_i$ for any $\mathbf{t} := (t_1, \cdots, t_n)$ which is a feasible solution of the optimization problem,

$$\min \sum_{i=1}^n t_i \quad \text{subject to} \quad \min_{\theta \in \mathcal{C}} \sum_{i=1}^n t_i KL(\nu_{\theta^*,i} \| \nu_{\theta,i}) \geq \log(1/2.4\delta).$$

Taking $\mathbf{t}^*$ to be an optimal solution to the previous problem, note that

$$\min_{\theta \in \mathcal{C}} \sum_{i=1}^n \frac{t_i^*}{\sum_{j=1}^n t_j^*} KL(\nu_{\theta^*,i} \| \nu_{\theta,i}) \geq \frac{\log(1/2.4\delta)}{\sum_{j=1}^n t_j^*} \geq \frac{\log(1/2.4\delta)}{\sum_{j=1}^n \mathbb{E}[T_j]}$$

In particular, since $\sum_{i=1}^n \frac{t_i^*}{\sum_{j=1}^n t_j^*} = 1$, we see that

$$\max_{\lambda \in \triangle_n} \min_{\theta \in \mathcal{C}} \sum_{i=1}^n \lambda_i KL(\nu_{\theta^*,i} \| \nu_{\theta,i}) \geq \frac{\log(1/2.4\delta)}{\sum_{i=1}^n \mathbb{E}[T_i]}.$$

Rearranging, we see that

$$\sum_{i=1}^n \mathbb{E}[T_i] \geq \log(1/2.4\delta) \min_{\lambda \in \triangle_n} \max_{\theta \in \mathcal{C}} \frac{1}{\sum_{i=1}^n \lambda_i KL(\nu_{\theta^*,i}||\nu_{\theta,i})}. \tag{6}$$

Now for $j \neq 1$, $\lambda \in \triangle^n$ and $\epsilon > 0$, define

$$\theta_j(\epsilon, \lambda) = \theta^* - \frac{(y_j^\top \theta^* + \epsilon)A(\lambda)^{-1}y_j}{y_j^\top A(\lambda)^{-1}y_j}.$$

where $A(\lambda) := \sum_{i=1}^n \lambda_i x_i x_i^\top$ and $y_j = z_1 - z_j$. Note that $y_j^\top \theta_j(\epsilon, \lambda) = -\epsilon < 0$ which implies that $\theta_j \in \mathcal{C}$. Also, the KL-divergence is given by

$$KL(\nu_{\theta^*,i}||\nu_{\theta_j(\epsilon,\lambda),i}) = (x_i^\top(\theta^* - \theta_j(\epsilon,\lambda)))^2$$

$$= y_j^\top A(\lambda)^{-1} \frac{(y_j^\top \theta^* + \epsilon)^2 x_i x_i^\top}{(y_j^\top A(\lambda)^{-1} y_j)^2} A(\lambda)^{-1} y_j.$$

Hence, returning to (6), we have that

$$\sum_{i=1}^n \mathbb{E}[T_i] \geq \log(1/2.4\delta) \min_{\lambda \in \triangle_n} \max_{\theta \in \mathcal{C}} \frac{1}{\sum_{i=1}^n \lambda_i KL(\nu_{\theta^*}||\nu_\theta)}$$

$$\geq \log(1/2.4\delta) \min_{\lambda \in \triangle_n} \max_{j=2,\cdots,m} \frac{1}{\sum_{i=1}^n \lambda_i KL(\nu_{\theta^*,i}||\nu_{\theta_j(\epsilon,\lambda),i})}$$

$$\geq \log(1/2.4\delta) \min_{\lambda \in \triangle_n} \max_{j=2,\cdots,m} \frac{(y_j^\top A(\lambda)^{-1}y_j)^2}{(y_j^\top \theta^* + \epsilon)^2 y_j^\top A(\lambda)^{-1}(\sum_{i=1}^n \lambda_i x_i x_i^\top)A(\lambda)^{-1}y_j}$$

$$= \log(1/2.4\delta) \min_{\lambda \in \triangle_n} \max_{y \in \mathcal{Y}^*(\mathcal{Z})} \frac{y_j^\top A(\lambda)^{-1}y_j}{(y_j^\top \theta^* + \epsilon)^2}$$

where in the second to last line we used the fact that $\sum_{i=1}^n \lambda_i x_i x_i^\top = A(\lambda)$. Letting $\epsilon \to 0$ establishes the result.

**Remark:** Note that $\theta_j = \text{argmin}_{\theta \in \mathbb{R}^d} ||\theta - \theta^*||_{A(\lambda)}^2$ subject to $y_j^\top \theta = -\epsilon$. $\qquad \square$

## D  Proof of Proposition 1

*Proof.* Assume $d$ is even and each $\epsilon_t \sim \mathcal{N}(0,1)$. Fix some $\alpha \in (0,1)$ which will depend on $\gamma$ in a clear way momentarily, and consider an instance where $\mathcal{X} = \mathcal{Z} = \{e_i\}_{i=1}^{d/2} \cup \{\cos(\alpha)e_i + \sin(\alpha)e_{d/2+i}\}_{i=1}^{d/2}\}$ where $e_i$ is the $i$-th standard basis vector.

If an algorithm is $\delta$-PAC, and takes $N_i$ samples from arm $i$, then for any $j \leq d/2$ it will be able to distinguish between $\theta = z_j$ and $\theta = z_{j+d/2}$. By standard Le Cam arguments [32] this hypothesis test requires $N_j + N_{j+d/2} \geq \frac{c\log(1/\delta)}{(1-\cos(\alpha))^2}$ for some universal constant $c > 0$. Because $(1-\cos(\alpha))^2 \approx \alpha^4/4$ and these inequalities must hold for all $j = 1, \ldots, d/2$ simultaneously for the single static allocation, we obtain the result. $\qquad \square$

## E  Proof of Lemma 1

*Proof.*

$$\rho(\mathcal{Y}) = \min_{\lambda \in \triangle_{|\mathcal{X}|}} \max_{y \in \mathcal{Y}} ||y||^2_{(\sum_{x \in \mathcal{X}} \lambda_x x x^\top)^{-1}}$$

$$= \frac{1}{\gamma_{\mathcal{Y}}^2} \min_{\lambda \in \triangle_{|\mathcal{X}|}} \max_{y \in \mathcal{Y}} ||y\gamma_{\mathcal{Y}}||^2_{(\sum_{x \in \mathcal{X}} \lambda_x x x^\top)^{-1}}$$

$$\leq \frac{1}{\gamma_{\mathcal{Y}}^2} \min_{\lambda \in \triangle_{|\mathcal{X}|}} \max_{x \in \text{conv}(\mathcal{X} \cup -\mathcal{X})} ||x||^2_{(\sum_{x \in \mathcal{X}} \lambda_x x x^\top)^{-1}}$$

$$= \frac{1}{\gamma_{\mathcal{Y}}^2} \min_{\lambda \in \triangle_{|\mathcal{X}|}} \max_{x \in \mathcal{X}} ||x||^2_{(\sum_{x \in \mathcal{X}} \lambda_x x x^\top)^{-1}}$$

The third equality follows from the fact that the maximum value of a convex function on a convex set must occur at a vertex. By the celebrated Kiefer-Wolfowitz theorem for G-optimal design [27], $\min_{\lambda \in \triangle_{|\mathcal{X}|}} \max_{x \in \mathcal{X}} \|x\|^2_{(\sum \lambda_x x x^\top)^{-1}} = d$ so we see that $\rho(\mathcal{Y}) \leq d/\gamma^2_{\mathcal{Y}}$. For a lower bound, note that

$$\min_{\lambda \in \triangle_{\mathcal{X}}} \max_{y \in \mathcal{Y}} \|y\|^2_{(\sum_{x \in \mathcal{X}} \lambda_x x x^\top)^{-1}} \geq \min_{\lambda \in \triangle_{\mathcal{X}}} \max_{y \in \mathcal{Y}} \sigma_{\min}((\sum_{x \in \mathcal{X}} \lambda_x x x^\top)^{-1}) \|y\|^2_2$$

$$= \min_{\lambda \in \triangle_{\mathcal{X}}} \max_{y \in \mathcal{Y}} \|y\|^2_2 / \sigma_{\max}((\sum_{x \in \mathcal{X}} \lambda_x x x^\top)^{-1})$$

where $\sigma_{\max}$ and $\sigma_{\min}$ are respectively the largest and smallest eigenvalue operators of a matrix. Since $\sigma_{\max}(\sum_{i=1}^n \lambda_i x_i x_i^\top) \leq \max_{x \in \mathcal{X}} \|x\|_2$, we have that $\rho(\mathcal{Y}(S_t)) \geq \max_{y \in \mathcal{Y}(S_t)} \|y\|^2_2 / (\max_{x \in \mathcal{X}} \|x\|_2)$. The final statement in the case of a singleton is also known as Elfving's Theorem, see Section 2.14 in [27] $\qquad\square$

## F  Experiment Details

In this section, we provide further details on the implementation of each algorithm. Each experiment was repeated 20 times with the mean sample complexity is reported and error bars representing the standard error are plotted. Simulations were implemented in Python 3 and parallelized on an Intel(R) Xeon(R) CPU E5-2690.

For each algorithm that requires computing a design $\lambda$ from an optimization of the form $\min_{\lambda \in \triangle_{\mathcal{X}}} \max_{s \in \mathcal{S}} \|s\|^2_{(\sum_x \lambda_x x x^\top)^{-1}}$ for $\mathcal{S} \subset \mathbb{R}^d$ (RAGE, $\mathcal{X}\mathcal{Y}$-static, $\mathcal{X}\mathcal{Y}$-oracle, and ALBA) we used a Franke-Wolfe algorithm [18] with constant step-size $2/(k+2)$ ($k$ being the iteration counter). The algorithm was run until the relative change in $\lambda$ with respect to the $\ell_2$ norm was less than .01 or 5000 iterations were reached. Any values of $\lambda < 10^{-5}$ were then thresholded to 0 and $\lambda$ was scaled to sum to 1.

- $\mathcal{X}\mathcal{Y}$-Adaptive [30]: This algorithm requires a parameter $\alpha$ that governs the length of each adaptive phase. We follow the simulations in [30] and let $\alpha = 0.1$. We remark that the algorithm given in the paper implements a greedy update to select arms in contrast to rounding the optimal allocation as is considered in the analysis. We implement the greedy arm selection procedure to match the simulations in the paper. It is worth noting that in several of the recent linear bandit papers that have implemented this algorithm, the active arm set has been reset at the conclusion of a phase before discarding arms. We do not reset the arm set at the conclusion of a phase to match what was done in [30]. Finally, in the confidence interval, we include the phase index and not the number of samples since we only need to union bound over when it is evaluated.

- $\mathcal{X}\mathcal{Y}$-Static and $\mathcal{X}\mathcal{Y}$-Oracle: To implement each allocation, we compute the optimal design on the set $\mathcal{Y}(\mathcal{Z})$ for the static strategy and the set $\mathcal{Y}^*(\mathcal{Z})$ normalized by the gaps for the oracle. Each algorithm is ran in phases in which $\gamma^t$ samples are drawn from the allocation. We experimented using $\gamma$ in the range $(1, 2)$ to optimize the performance of the algorithms and ended up using $\gamma = 1.1$ for the oracle strategy and $\gamma = 1.35$ for the static strategy across the examples. The stopping condition $\{\exists z' \in \mathcal{Z} | \forall z \in \mathcal{Z} : \|z' - z\|_{A_t^{-1}} \sqrt{2 \log(2t^2 |\mathcal{Z}|^2/\delta)} \leq (z' - z)^\top \widehat{\theta}_t\}$ is evaluated at the end of each phase $t$ to decide when to terminate the experiment for each algorithm, but only union bounding over $|\mathcal{Z}|$ for the oracle.

- LinGapE [33]: We run this algorithm with a regularizer on the least squares estimator of $\lambda = 1$ following the implementation given in the paper. LinGapE is designed to find an $\varepsilon$ good arm. We let $\varepsilon = 0$ to ensure the optimal arm is identified. The simulations in [33] apply a greedy arm selection strategy that deviates from the algorithm that is analyzed. We instead implement the LinGapE algorithm in the form that it is analyzed.

- ALBA [31]: This algorithm is parameter free and we implement the $\mathcal{Y}$-ElimTil sub-procedure following the paper since it gives improved empirical results compared to the $\mathcal{X}$-ElimTil sub-procedure that provides identical theoretical results.

- RAGE: To compute the discrete allocation given a design, we use the rounding procedure discussed in Section B from [27] and $\epsilon = 1/10$. The algorithm we propose is computationally efficient since there is at most $\lfloor \log_2(4/\Delta_{\min}) \rfloor$ phases and each phase only requires solving a convex optimization to obtain the design, an efficient rounding procedure, and solving a least squares problem. The time required between each pull is negligible.