[Reviews · NeurIPS 2019]

Reviewer 1



The paper introduces a natural yet interesting extension to the pure exploration in linear stochastic bandits, called pure exploration in transductive linear bandits. This extension is well motivated through various examples in the introduction, which makes the paper pleasant to read. Overall the paper is very well written and is a nice read. It is very well organized, and almost error free. The studied problem is presented precisely and clearly. Moreover, the steps towards the design of the algorithm are described and motivated very clearly. In addition, the numerical experiments are explained and detailed very well. All these make the paper a nice read. From the technical side, a positive aspect of the paper is that it provides a complete picture of the introduced problem: the problem-dependent lower bound on the sample complexity, as well as an algorithm whose performance matches the lower bound (up to logarithmic factors). I read the proofs (without checking all the details), and believe they are flawless and correct. Detailed comments. 1. As far as I understood, both lower and upper bounds follow straightforwardly from by-now-standard proof techniques from the previous analyses for linear bandits in the literature. Hence, despite the clear presentation of the paper and matching upper and lower bounds provided, given the very high standard of NeurIPS, I think the aforementioned contributions do not make the paper strong enough to pass the acceptance threshold. Therefore, I may ask the authors to highlight the main non-trivial steps in their proofs, which makes the proof challenging/or would call for new technical tools compared to existing works. 2. What is the effect of choosing d odd in Proposition 1? 3. Line 191: … Algorithm 1 matches the lower bound … --> the sample complexity of Algorithm 1 matches … --- Updates after rebuttal --- I have gone through the authors' response. I appreciate their response to my comment. My impression has been improved, and I increase my score to 6. Still I think the described contribution does not strongly justfiy publication at NeurIPS. Additional comment: Line 233: where $i$ is ... ==> I think you meant "where $\Delta_i$ is ..."

Reviewer 2



The paper proposes the general problem setting of the pure exploration for transductive linear bandits, in which a set of arms in X could be used for exploration, and the reward of an arm x (a d-dimensional vector) is the dot-product of x with an unknown latent vector \theta^*, and the reward is observed after playing arm x. The goal, however, is to find the optimal arm z from a different set of arms Z, with the same reward definition. The authors motivate the problem from applications such as drug discovery, recommendations etc. The formulation generalizes the linear bandit case (where X = Z) and the combinatorial bandit case, where X is the set of d standard basis vectors. The problem looks interesting and useful. The authors design an algorithm SAGE, which combines existing techniques such as gap elimination and rounding procedure. The authors claim that the algorithm is better than existing algorithms, for example comparing against ALBA of [28], they claim that ALBA does not compute optimal allocation based on differences but on arms in X directly, which is not as efficient. The authors provide a sample complexity upper bound for SAGE and a corresponding lower bound theorem. The upper bound looks complicated, but the authors do try to provide interpretations of the upper bound result. The authors conduct experiments comparing SAGE with several linear bandit algorithms. The results show that SAGE is competitive, and has advantages in various settings. Overall, the paper provides a nice and useful general setting of the transductive linear bandit problem, and provide both analytical and empirical results to the problem. The paper overall is well organized and well written, with balanced intuitive explanations and technical results. I think the paper is worth to be published. However, there are still some (minor) issues, as listed below. - The paper never formally define what is an arm. In some papers on linear bandit, an arm is referred to as one dimension in the action vector, while in others an arm is the vector itself. Thus I believe a clarification is needed here. I believe in this paper an arm is an vector, but there are two sets of vectors X and Z. Are vectors in both sets called arms? This actually leads to another issue below. - Line 82, $\sum_{x\in X} (\lambda_x x x^\top)^{-1}$ first appears in this line, but there is no explanation about $\lambda_x x x^\top$. For example, is it full-rank? What if it is not? I only find a relevant explanation in Footnote 2 two pages later. Please clarify this point up front. - Algorithm 1. a) The computation of \lambda^*_t needs more explanation. How efficient is it? Only in the experimental section the authors mention that they use Frank-Wolfe algorithm to find \lambda^*_t. I hope more explanation could be given in the algorithm section. b) What is n_\min? Should it be r(\epsilon)? c) What is \lambda^* in the ROUND procedure? Should it be \lambda^*_t? - Experiments. a) The transductive setting. The setting used is actually the pure exploration for combinatorial bandit setting. Then baselines for this setting should be compared. Moreover, a separate setting for the general transductive setting (neither the linear bandit setting nor the combinatorial bandit setting) should be tested. b) The sub-Gaussian noise setting is not mentioned. What is the parameter used? - Proof of Theorem 1 in the supplementary material. In line 488, it explicitly defines \nu_{\theta,i} as the normal distribution with mean z_i^{\top} \theta and variance 1. However, in the formula after line 497, x_i^{\top} appears, which really confuses me. Should this x_i be z_i, or the other way round --- the z_i mentioned above should be x_i? Another indication is that the summations in the previous formulas are all from 1 to n, suggesting that this is going through vectors in X, not in Z. Please clarify this. This is also related to the first issue I listed. For example, in line 487, arm i is mentioned, but is it arm i in X or Z? The differentiation between X and Z is crucial in the transductive setting, but it seems the differentiation is unclear in the proof of Theorem 1. Please check this differention in all the proofs, since I did not check all the details of all the proofs.

Reviewer 3



-------Comments added after the rebuttal------------- 1. Prop 1: Thanks for clarifying the implication of proposition 1. Please add a discussion to the paper. 2. The differences between the implemented algorithm in [1] and the algorithm in this paper should be clearly mentioned in the paper. " slightly different condition to discard candidate vector" -- it would be better to be specific and add the details. ----------------------------------------- I think this is a good and well written paper. There are several interesting results (see above). There were few issues with the writing and I have some general questions. 1. In the abstract and in several places in the paper it has been written that "this is a generalization of linear and combinatorial bandits". It would be better if it is clarified everywhere that this is only the pure exploration version to avoid confusion. 2. (not a drawback) I am curious whether the bounds derived in this paper can lead to any new insight or better problem dependent bound for the cumulative regret problem in linear bandits. 3. It would be better to give direct references to the results used in the "review of least-squares" section even though the results are fairly well-known. 4. I actually did not understand the significance of proposition 1. What does it justify? A bit more discussion will be helpful. 5. Can you clarify whether the only difference between the version of [27] used in the empirical section and RAGE is the greedy rounding procedure? in light of "We remark here that in our implementation of the X Y -Adaptive allocation, we follow 267 the experiments in [27] and allow for provably suboptimal arms to be discarded (though this is not 268 how the algorithm is written in their paper). " 6. It would be great if a real dataset or experiment can be identified to try these algorithms out. It would be great to see the performances of the various algorithms where the linearity assumptions are not exactly satisfied. For example one can find the optimal parameter from a training set in one of the datasets used to evaluate linear bandits. Then on a separate validation set, these algorithms can be tested on whether they can identify the above parameter.

Reviewer 4



I find the new transductive setup well-motivated and interesting. Still, I wonder if this setting is really harder than pure exploration in linear bandits. Indeed, the lower bound derivation (and the complexity quantity) follow very closely the one obtained in the linear case. This makes me wonder whether any algorithm for pure-exploration in linear bandit could be extended to best arm identification in the transductive setup. Still, the authors propose a specific algorithms that is not just an extension of an existing one (at least not that I noticed), and perform an original analysis (that I did not check in details). The main theoretical claim of the paper is that in the linear setting, where several pure exploration algorithms have recently been proposed, they achieve proposed a sample complexity upper bound that is the closest to the lower bound. My only concern about the paper is that I am not completely sure about this claim. Indeed, the paper [30] kind of claim the same thing. Yet in Section 4, the upper bound obtained in this work is not even mentioned: the LinGapE algorithm is "ruled out" of the pool of good algorithms because it needs to select each arm once. Still, it could be that in a regime of a moderate number of arms, the upper bound obtained in [30] is better than the one obtained in the present paper. I agree that the upper bound in [30] is less explicitly related to the lower bound than the one in the present paper, however a fair comparison would for example numerically compute the two upper bounds to juge which one is closer to the lower bound. In Theorem 2 we have a multiplicative factor of c * \ln(1/\Delta_\min). First, it would be nice to specify the value of c in the statement. Then, \ln(1/\Delta_\min) can actually be very large, especially with 15000+ randomly chosen arms in dimension 2 as in the experiment reported in Figure 1(b). To summarize, without the ln(1/\D_\min) this paper would be a strong accept due to the new framework + first optimal algorithm for pure exploration in linear bandit. Without it, the reader needs to be further convinced of the actual improvement in the theoretical result for the linear case.

[Author Response · NeurIPS 2019]

We thank the reviewers for providing detailed and quality reviews. The primary objective of this paper is to provide the first linear bandit algorithm with a sample complexity nearly matching the information-theoretic lower bound.

**Reviewer 2:** We thank the reviewer for the positive comments and encouraging us to point out the novelty in our analysis. Despite extensive work on the fixed-confidence pure exploration linear bandit problem dating back to 2014, no existing paper has presented a near-optimal non-asymptotic algorithm. In the most recent work on linear bandits [2], obtaining such a result was raised as an existing open problem in the conclusion and our work gives a clear answer. We present the first non-asymptotic algorithm that nearly achieves the information-theoretic lower bound. Obtaining this result necessarily requires insights into the problem and analysis techniques beyond what has appeared in the literature.

The primary novelty in our analysis is quantifying the relationship between the algorithm sample complexity and the lower bound we provide. We characterize this relationship in the second inequality following line 438 and it is proven in the analysis that follows line 439. In this portion of the proof we show that the algorithm sample complexity including the problem-dependent terms matches the lower bound up to logarithmic factors. No analysis of this type has appeared in the literature previously and it is the fundamental component in the proof to show the algorithm we present is nearly optimal. This analysis opens up future research avenues to refine the proof technique in an effort to remove the $\log(1/\Delta_{\min})$ term showing up in the final result. A number of recent results on linear bandits have appeared, yet since they were unable to relate the sample complexity bounds to the lower bound as we do in our work, it has not been clear if any of them have come close to matching the lower bound. As we discuss in the related work section and demonstrate in our numerical experiments, it turns out that several recent algorithms have fundamental flaws that may be challenging to notice since the bounds cannot be compared to the lower bound.

Unlike the upper bound, we agree that the lower bound is obtained using standard techniques and we acknowledge this fact in our manuscript. However, we believe the tight lower bound we give is an important result to be able to characterize optimality. A final novelty in our analysis is obtaining tighter bounds on the problem-dependent terms. In Lemma 1, we show that $\rho(\mathcal{Y}) \leq d/\gamma_{\mathcal{Y}}^2$ where $\gamma_{\mathcal{Y}}$ is the gauge of $\mathcal{Y}$. In previous works [1], the problem-dependent quantity has been naively bounded as $\rho(\mathcal{Y}) \leq 4d$. The example after Lemma 1 shows the bound we provide can be significantly tighter. Proposition 1 is a lower bound, so $d$ being even in the statement is simply giving an example to show there is a problem instance for which a static strategy must incur a factor of $d$ samples more than necessary.

**Reviewer 3:** Thanks for pointing out the importance of clearly presenting the definition of an arm. In the transductive linear bandit problem, there are two finite sets of vectors $\mathcal{X} \subset \mathbb{R}^d$ and $\mathcal{Z} \subset \mathbb{R}^d$. An arm is a $d$-dimensional vector in the set $\mathcal{X}$ that can be measured directly, while vectors in the set $\mathcal{Z}$ cannot be measured directly. The objective is to identify the vector $z^* = \arg\max_{z \in \mathcal{Z}} z^\top \theta^*$ while obtaining only measurements from the set of arms of the form $x^\top \theta^* + \eta$.

The transductive experiment at the end of the experiments section is a general transductive example and not a combinatorial bandit problem since the set $\mathcal{Z} \not\subset \{0,1\}^d$. Comparing to existing combinatorial bandit algorithms is an interesting direction of future work. In the experiments the noise was generated from a standard normal distribution; this detail was included in the appendix on line 522, but we plan to move it to the main paper.

We appreciate you pointing out several typos that we are fixing. In Algorithm 1, $n_{\min}$ should be $r(\epsilon)$ and $\lambda^*$ should be $\lambda_t^*$ as you pointed out. In general, when $\sum_{x \in \mathcal{X}} \lambda_x x x^\top$ is non-invertible, pseudo-inverse can be applied. We will provide further details on the computation of $\lambda^*$ in the paper thanks to your suggestion. While the computationally complexity of solving for $\lambda^*$ is not a primary focus of our paper, empirically we find that it can be obtained efficiently.

Thanks for the detailed comments on the proof of Theorem 1. There is a typo on line 488 and $\nu_{\theta,i}$ should be defined as the reward distribution of the $i$–th arm in $\mathcal{X}$ so that $\nu_{\theta,i} = \mathcal{N}(x_i^\top \theta, 1)$. The index $i$ is with respect to the $n$ elements in $\mathcal{X}$ and the index $j$ is with respect to the $m$ elements in $\mathcal{Z}$.

**Reviewer 4:** Thank you for the careful review and giving several useful suggestions. We will make it clear the generalizations pertain only to pure exploration. Exploring insights into problem-dependent regret bounds is an interesting direction for future work. We plan to edit the paper to include pertinent references to the results in the least squares section. Proposition 1 says that there exists a problem instance for which a static strategy must incur a factor of $d$ samples more than the optimal sample complexity. This indicates that it is necessary to devise an adaptive algorithm to obtain a near-optimal sample complexity. We will clear this up in the paper. The version of the algorithm from [1] implemented in the experiments is similar to our algorithm. The primary discrepancies are the greedy rounding procedure, phases being of a random length, and a slightly different condition to discard candidate vectors.

[1] Marta Soare, Alessandro Lazaric, and Rémi Munos. Best-arm identification in linear bandits. In *Advances in Neural Information Processing Systems*, pages 828–836, 2014.

[2] Chao Tao, Saúl Blanco, and Yuan Zhou. Best arm identification in linear bandits with linear dimension dependency. In *International Conference on Machine Learning*, pages 4884–4893, 2018.


[Meta-Review · NeurIPS 2019]

All the reviewers saw some value in the submission, but also the some unanswered questions that should be addressed. An additional expert reviewer was added as a result. Based on the 4 reviews and the discussion, I suggest acceptance, in expectation that the authors address as much as possible the issues raised in the reviews.